# A Comparative Analysis of Influenza-Associated Disease Burden with Different Influenza Vaccination Strategies for the Elderly Population in South Korea

**DOI:** 10.3390/vaccines10091387

**Published:** 2022-08-25

**Authors:** Min Joo Choi, Jae-Won Yun, Joon Young Song, Karam Ko, Joaquin F. Mould, Hee Jin Cheong

**Affiliations:** 1Department of Internal Medicine, International St. Mary’s Hospital, Catholic Kwandong University College of Medicine, Incheon 22711, Korea; 2Asia Pacific Influenza Institute, Korea University College of Medicine, Seoul 08308, Korea; 3Department of Internal Medicine, Guro Hospital, Korea University College of Medicine, Seoul 08308, Korea; 4Seqirus Korea Ltd., Seoul 03157, Korea; 5Seqirus USA Inc., Summit, NJ 07901, USA

**Keywords:** influenza, disease burden, averted outcomes, quadrivalent vaccine, elderly

## Abstract

Influenza affects all age groups, but the risk of hospitalization and death due to influenza is strongly age-related and is at its highest among the elderly aged 65 years and older. The objective of this study is to compare the differences in influenza-associated disease burden under three different influenza vaccination strategies—the standard-dose quadrivalent influenza vaccine (QIV), high-dose QIV (HD-QIV), and MF59^®^-adjuvanted QIV (aQIV)—for the elderly population aged 65 years and older in South Korea. A one-year decision-tree model was developed to compare influenza disease burdens. The input data for the model were obtained from published literature reviews and surveillance data from the Korea Disease Control and Prevention Agency (KDCA). The analysis indicated that aQIV is more effective than QIV, preventing 35,390 influenza cases, 1602 influenza-associated complications, 709 influenza-associated hospitalizations, and 145 influenza-associated deaths annually. Additionally, aQIV, when compared to HD-QIV, also reduced the influenza-associated burden of disease, preventing 7247 influenza cases, 328 influenza-associated complications, 145 influenza-associated hospitalizations, and 30 influenza-associated deaths annually. Switching the vaccination strategy from QIV to aQIV is predicted to reduce the influenza-associated disease burden for the elderly in South Korea. The public health gains from aQIV and HD-QIV are expected to be comparable. Future studies comparing the effectiveness of the vaccines will further inform future vaccination strategies for the elderly in South Korea.

## 1. Introduction

Every year, seasonal influenza leads to a significant public health burden. According to a dataset that complied the statistics of WHO regions consisting of 31 countries from 2002 to 2011, the global influenza-associated excess mortality rate is estimated to be 5.9 per 100,000 people on average each year [1]. Similarly, in South Korea, the influenza-associated excess mortality rate from 2003 to 2013 was found to be 5.97 per 100,000 people annually [2]. Influenza affects all age groups, but the risk of hospitalization and death due to influenza is strongly age-related and is at its highest among the elderly aged 65 years and older. Globally, the population aged ≥65 years accounts for 58% of all influenza-associated respiratory deaths every year [3]. Although the vaccination coverage among those populations in South Korea is over 80%, the burden of influenza in this specific age group is still significant. It is estimated that the influenza-associated excess mortality rate of the elderly population (≥65 years) is 46.98 per 100,000, which is almost eight times the influenza-associated excess mortality rate of all populations in South Korea [2]. One of the attributable factors for this is age-related immunosenescence [4]; immunosenescence leads to suboptimal vaccine-induced B cell and T cell-mediated adaptive immune responses and contributes to the reduced effectiveness of influenza vaccines in the elderly population [5].

The influenza vaccination is widely recognized as the safest and most effective measure of preventing infections and severe outcomes caused by influenza viruses. Most human infections are caused by influenza A and influenza B viruses, whose circulation varies significantly by location and by season [6]. Standard-dose quadrivalent influenza vaccines (QIV) have been developed to cover four different virus subtypes, namely: A(H1N1) and A(H3N2), as well as the B/Victoria and B/Yamagata lineages [7]. Besides QIV, several enhanced vaccines, including MF59^®^-adjuvanted QIV (aQIV) and high-dose QIV (HD-QIV), have been developed and approved in multiple countries to improve vaccine effectiveness in the elderly [8,9,10]. aQIV contains MF59^®^, an oil-in-water emulsion adjuvant, and is designed to improve the magnitude, breadth, and persistence of the immune response compared to non-adjuvanted influenza vaccines [11]. HD-QIV contains four times the dose antigen (the part of the vaccine that helps patients build up protection against influenza viruses) [10]. As of the 2020–2021 season, all influenza vaccines in the South Korean National Immunization Program (NIP) are standard-dose quadrivalent vaccines (QIV) [12], and thus, this study will focus on the comparisons among the three quadrivalent vaccines.

This study aims to compare the estimated influenza-associated disease burden for the elderly in South Korea under three different influenza vaccination strategies (QIV, HD-QIV, and aQIV). Even though neither HD-QIV nor aQIV has been approved in South Korea as of July 2022, this study may provide a foundation for the South Korean Government to broaden influenza vaccination options by comparing the influenza-related burden averted using three different vaccination strategies, once these vaccines become available in South Korea.

## 2. Materials and Methods

### 2.1. Model Design and Structure

We used a static one-year decision tree model from the perspective of the South Korean healthcare system for this analysis. Our model aimed to compare influenza-associated disease burdens under three influenza vaccination strategies (Figure 1). The influenza burden was explicitly modeled for three elderly patient cohorts defined by age (65–74, 75–84, and ≥85 years). The intervention under consideration is the South Korean NIP for the elderly using aQIV. In this study, the NIPs for the same population using QIV or HD-QIV were chosen as comparators. The model outcomes are the total number of influenza cases and influenza-associated complications, hospitalizations, and deaths under each intervention.

Upon entering the decision-tree model, each cohort was further divided into high-risk and low-risk subgroups based on the comorbidity distribution previously reported in a South Korean influenza modeling study [13]. High-risk patients were defined as a population with one or more of the following underlying conditions: chronic respiratory disease, chronic heart disease, chronic renal disease, chronic liver disease, neurological disease, metabolic syndrome, malignancy, hematologic disease, or an immunocompromised state. Low-risk patients were defined as a population without the conditions mentioned above. The model follows each patient group and computes their corresponding influenza cases [13]. Influenza-associated hospitalizations were estimated for patients with influenza based on their complication status. Using the previously published data in South Korea [13], the probabilities of inpatient and outpatient influenza-associated death were calculated.

### 2.2. Model Inputs

#### 2.2.1. Demographic Inputs

According to government statistics, in 2017, there was an elderly population of 8.1 million in South Korea, of which 4.6 million, 2.7 million, and 0.8 million were in the 65–74, 75–84, and ≥85-year age groups, respectively [14]. The comorbidity risk is 16.64%, 23.57%, and 25.71%, respectively, for each age group mentioned above [15]. Consistent with a previously published South Korea model [13], a vaccination coverage rate of 80% for the elderly population was used in the base-case calculation.

#### 2.2.2. Vaccine Effectiveness Inputs

The absolute vaccine effectiveness (VE) was obtained from a published international meta-analysis. In the base-case analysis, the VEs and 95% confidence intervals (CIs) of QIV for the elderly population against A(H1N1), A(H3N2), and B were 62% (36% to 78%), 24% (−6% to 45%), and 63% (33% to 79%), respectively [16]. Consistent with the practice adopted by the European Medicines Agency [17], the relative vaccine benefits observed from the MF59^®^-adjuvanted trivalent vaccine are inferred for aQIV because both vaccines are manufactured using the same process and have overlapping compositions. The mean relative vaccine effectiveness (rVE) of aQIV vs. QIV was 13.9% (95% CI: 4.2% to 23.5%) and the rVE of aQIV vs. HD-QIV was 3.2% (95% CI: −2.5% to 8.9%), the same as those reported by Coleman et al. for the trivalent influenza vaccines [18].

#### 2.2.3. Influenza Epidemiological Inputs

Data on the circulation of influenza virus types and subtypes were obtained from the KDCA nationwide surveillance data between 2016 and 2019 [19,20]. The circulation data for the season between 2019 and 2021 were excluded to avoid potential bias due to the coronavirus disease 2019 (COVID-19) pandemic. It is hard to accurately predict the disease burden of influenza during the pandemic as the non-pharmacological interventions could lead to a reduction in influenza outbreaks and disease burden, and co-infection with the COVID-19 could lead to an increase in disease burden [21,22,23]. Influenza virus type and subtype distributions are shown in the Appendix A. For the base-case analysis, the average distribution from 2016–2019 was used: 16.5% for A (H1N1), 44.0% for A (H3N2), and 39.5% for B viruses.

There is no published attack rate of seasonal influenza among the unvaccinated available for South Korea; thus, the model assumes the annual seasonal influenza attack rate for the healthy unvaccinated population to be 7.2%, based on a systematic literature review with meta-analysis [24]. The attack rate for the healthy unvaccinated population was used to calculate the influenza incidence for the vaccinated with the vaccine effectiveness of the three different influenza vaccines. The strain-specific influenza attack rates were calculated by multiplying the attack rate by the strain’s prevalence [24]. The risks of influenza-associated complications and hospitalizations, stratified by age group and commodity risk level, are reported in Table 1. Mortality risks stratified by age group, risk level, complication, and hospitalization status are reported in Table 2.

### 2.3. Sensitivity Analysis

One-way deterministic sensitivity analysis (DSA) and probabilistic sensitivity analysis (PSA) were performed to test the stability of the model results under the uncertainty of the input variables. In the DSA, the minimum and maximum values of each parameter were tested, one at a time, in the simulation, and a tornado diagram was used to assess which of the model’s parameters has the greatest influence on its results [25]. In PSA, values for all the parameters were randomly drawn from the corresponding distribution at the same time. For each comparison, a total of 1000 PSA Monte Carlo simulation runs were performed.

When 95% CIs were available, they were used as the minimum and maximum values in the DSA. When 95% CIs were not reported, a ±10% range was used to define the minimum and maximum values. Beta-distributions were assumed in the PSA for all the probability parameters. Normal distributions were assumed for the others. The details of the variables used in sensitivity analysis, including the base-case value, minimum and maximum values, and distributions, are reported in Appendix A.

### 2.4. Number Needed to Vaccinate (NNV)

In addition to the influenza disease burden comparison using the vaccine effectiveness and epidemiological inputs, another common option for comparing different vaccination strategies is to use the number needed to vaccinate (NNV). The NNVs were calculated based on the strain-specific vaccination effectiveness and influenza attack rates utilized in our investigation [26], where:

NNV = 1/(annual incidence in the unvaccinated × vaccine effectiveness)

The annual incidence in the unvaccinated was assumed to be 7.2% [24], and vaccine effectiveness for the three different QIVs was adopted from the meta-analyses [16]. The strain-specific influenza incidence rate was calculated by multiplying the overall attack rate by the strain’s prevalence [24].

## 3. Results

In the base-case scenario, the influenza disease burden from the NIP with aQIV was compared against the NIP with QIV or HD-QIV. When aQIV was compared with QIV, the model calculation indicated a 9.52% reduction in total influenza cases, which translates into 35,390 fewer influenza cases, 1602 fewer influenza-associated complications, 709 fewer influenza-associated hospitalizations, and 145 fewer influenza-associated deaths in South Korea (Table 3).

In addition, the calculation indicated that HD-QIV, compared to QIV, reduced the total influenza cases and associated disease burden by 7.57%. The inclusion of HD-QIV in the NIP reduced influenza cases by 28,143, influenza-associated complications by 1274, hospitalizations by 564, and deaths by 116, compared to QIV.

In terms of effectiveness, HD-QIV and aQIV were similar. The outcome comparisons between aQIV and HD-QIV in the NIP for the elderly population demonstrated that aQIV only has marginally better outcomes. The overall outcome was improved by 2.11%; aQIV reduced influenza cases by 7247. Influenza-associated complications and hospitalizations were lowered by 328 and 145, respectively. aQIV reduced influenza-associated deaths by 30 (Table 3).

DSAs were carried out independently for aQIV vs. QIV and aQIV vs. HD-QIV comparisons. The tornado charts display the top ten most influential factors for influenza cases in both comparisons (Figure 2 and Figure 3). The DSA results showed similar variable rankings in the two comparisons. It is not surprising to see that relative vaccine effectiveness is the most important factor when two vaccines are compared head-to-head. The influenza incidence rate for the unvaccinated, low-risk population is the second most influential variable in both comparative analyses. The DSA variable ranking patterns were consistent when other influenza-associated burdens were used as outcome variables in the analysis. The results of the additional DSA analyses are included in the Appendix A.

Probabilistic sensitivity analyses were also performed to test the stability of the base-case results against fluctuations in the input variables. Two 1000-round simulations for all the outcomes were performed separately: one for aQIV vs. QIV and the other for aQIV vs. HD-QIV. The PSA results further confirmed that using aQIV to replace QIV in the NIP for the elderly population would consistently reduce influenza cases and other influenza-associated burdens (Table 4). When using aQIV to replace HD-QIV in the same vaccination program, 86% of the simulations showed that aQIV outperformed HD-QIV. The results of the PSA analyses are included in the Appendix A.

Table 5 shows the appropriate NNVs based on the strain-specific vaccination effectiveness and influenza attack rates utilized in our investigation. QIV required a greater number of people to be vaccinated to prevent one influenza case than aQIV and HD-QIV for all the influenza subtypes, and the difference in NNV between aQIV and HD-QIV was marginal (Table 5).

## 4. Discussion

This study evaluates the disease burden of influenza from various influenza vaccination strategies for the elderly population in South Korea. Based on the currently available data, substituting QIV with aQIV in the South Korean National Immunization Program would be more effective in reducing the disease burden from seasonal influenza. For the same elderly population, aQIV also slightly outperforms HD-QIV. Both univariate and probabilistic sensitivity analyses confirmed that the above findings are resistant to changes in the input variables.

The findings of this study are consistent with prior studies on the public health benefits of an enhanced influenza vaccine for the elderly population in South Korea [13,27]. One study published in 2019 used similar demographic input data in their lifetime Markov model to compare the cost-effectiveness of adjuvanted trivalent influenza vaccine (aTIV) vs. standard TIV [13]. They calculated that aTIV outperformed TIV in reducing influenza cases by nearly 16% over a lifetime, resulting in a switch to aTIV as a cost-saving strategy despite being more expensive. In our one-year evaluation, we saw a similar impact on disease burden (aQIV vs. QIV), with a 9.52% reduction in influenza cases with aQIV. The relatively lower improvement in our analysis may be due to our more conservative rVE value for aQIV vs. QIV (13.9% in our study compared to 25% in the other [13]). A more recent study compared the cost-effectiveness of standard QIV vs. aTIV. They reported that aTIV reduced total influenza cases by 16.2% and was more cost-effective compared to the standard QIV, despite aTIV being a trivalent formulation and more expensive [27]. Studies conducted outside South Korea showed similar results. Cost-effectiveness analyses reported from the UK, Italy, and Argentina and the systematic review showed that from the public health perspective, the MF59^®^-adjuvanted influenza vaccine generally provided more public health benefits than the standard TIV/QIV, despite being more expensive [28,29,30,31].

Moreover, the NNV is a measure that highlights the potential benefits of the vaccination program. The NNV calculation findings in our study are within the previously reported range of NNVs for influenza vaccinations, which is 43 (95% CI: 16–192) [26]. Across all strains, QIV required a greater number of people to be vaccinated to prevent one influenza case, followed by HD-QIV and aQIV. Both the influenza disease burden outcomes and NNV results support the clinical value of aQIV and HD-QIV in the elderly.

Here, we reported the results from South Korea’s first epidemiological modeling study of three different QIVs. According to our findings, aQIV may be able to provide more public health benefits than QIV for the same population in South Korea, reducing more overall influenza cases, by 9.52%, compared to QIV. The proposed mechanism of potential benefit in which MF59^®^ helps build a broader immune response while prolonging its effectiveness may support the findings of our study [11]. While both HD-QIV and aQIV are intended for the elderly, our analysis suggests that aQIV is more likely to reduce the influenza-associated disease burden than HD-QIV. However, the result should be interpreted with caution as the meta-analysis [26] found only slightly higher vaccine effectiveness in aQIV than in HD-QIV, and the difference was not statistically significant. The relevant vaccine effectiveness is the most important variable influencing our study result. Thus, we interpret the minor disease burden difference between aQIV and HD-QIV as non-material. Nevertheless, the findings are stable against fluctuations in the input variables. When developing and implementing a national influenza vaccination plan in South Korea, it is critical to consider the cost and effectiveness of the approved quadrivalent vaccines. Once the costs for these upcoming vaccines are available, our modeling paradigm in this study can quickly be adapted to cost-effectiveness analysis.

There are several limitations to this study. First, our model is a one-year statical model that does not account for herd immunity or changes in transmission patterns when the vaccination rate reaches 80%. As a result, these disease burden reduction estimates may be overly conservative. To explicitly consider herd immunity, a dynamic model, such as an agent-based simulation, may be required. A dynamic simulation model, on the other hand, will necessitate far more input data than what is currently available in South Korea. Second, the impact of seasonal variations on disease burden might not be sufficiently reflected, with the NHIS claiming data from only two seasons (2013–2015). To overcome this potential shortcoming, seasonal variation was further assessed in the sensitivity analysis. Last, there is still very little information available about the rVE of quadrivalent influenza vaccines for the elderly population in South Korea. Local clinical evidence is needed to better understand the relative effectiveness of these vaccines.

## 5. Conclusions

Influenza remains one of the major public health challenges, leading to hospitalizations and mortality in the elderly. Enhanced vaccines (aQIV and HD-QIV) could provide better protection than standard-dose QIV for the elderly. Based on this study, switching the vaccination strategy from QIV to aQIV is predicted to beneficially reduce the influenza-associated disease burden of the elderly population in South Korea. The public health gains from aQIV and HD-QIV are expected to be comparable. Once the cost of these vaccines becomes available, additional studies comparing the effectiveness of the QIVs will further inform future public vaccination strategies in South Korea.

## Figures and Tables

**Figure 1 vaccines-10-01387-f001:**
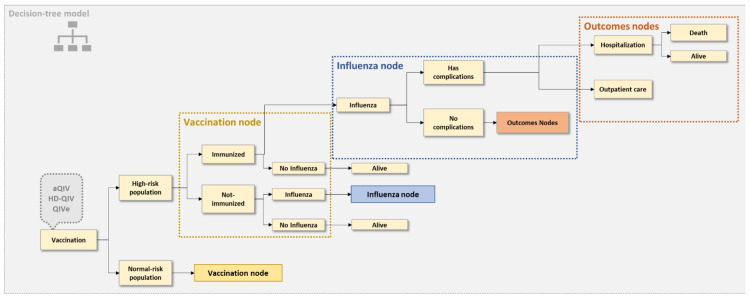
Model structure. aQIV—MF59^®^-adjuvanted quadrivalent influenza vaccine; HD-QIV—high-dose quadrivalent influenza vaccine; QIV—standard quadrivalent influenza vaccine.

**Figure 2 vaccines-10-01387-f002:**
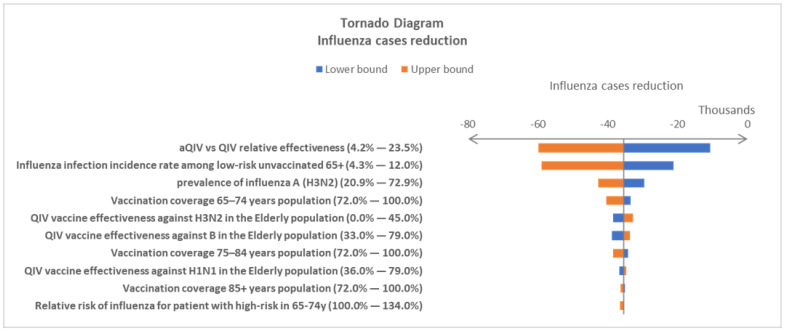
Top 10 most influential factors driving influenza case reduction in the comparison of adopting aQIV vs. QIV in the National Immunization Program (NIP) for the elderly population in South Korea. The values in parenthesis are the minimum and maximum values tested in the deterministic sensitivity analysis (DSA). aQIV—MF59^®^-adjuvanted quadrivalent influenza vaccine; QIV—standard quadrivalent influenza vaccine.

**Figure 3 vaccines-10-01387-f003:**
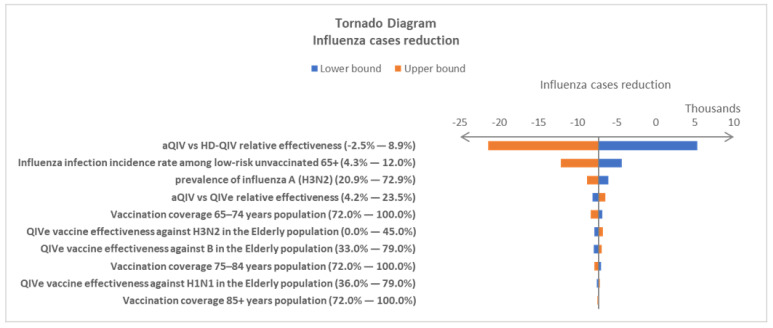
Top 10 most influential factors driving influenza case reduction in the comparison of adopting aQIV vs. HD-QIV in the National Immunization Program (NIP) for the elderly population in South Korea. The values in parenthesis are the minimum and maximum values tested in the deterministic sensitivity analysis (DSA). aQIV—MF59^®^-adjuvanted quadrivalent influenza vaccine; HD-QIV—high-dose quadrivalent influenza vaccine; QIV—standard quadrivalent influenza vaccine.

**Table 1 vaccines-10-01387-t001:** Complication and hospitalization percentages, stratified by age group and comorbidity risk level [13].

Outcomes	Age Group	Low Risk	High Risk
Complication percentage per influenza case	65–74 years	3.19%	7.78%
	75–84 years	2.94%	10.26%
	85+ years	3.97%	17.41%
Hospitalization percentage per influenza case	65–74 years	0.55%	5.03%
	75–84 years	0.74%	7.86%
	85+ years	1.37%	14.65%

**Table 2 vaccines-10-01387-t002:** Mortality stratified by age group, risk level, influenza-associated complication status, and hospitalization status [13].

Risk Level	Influenza-Associated Complication Status	Hospitalization Status	Mortality by Age Group		
			65–74 years	75–84 years	≥85 years
Low	Without complications	Outpatient	0.0003%	0.1115%	0.6477%
		Hospitalization	0.2731%	0.3257%	1.7506%
	With complications	Outpatient	0.0683%	0.2346%	1.6850%
		Hospitalization	1.2972%	2.0993%	7.3140%
High	Without complications	Outpatient	0.3813%	1.0777%	3.3455%
		Hospitalization	2.4090%	3.2067%	5.9396%
	With complications	Outpatient	1.6242%	3.3283%	8.7282%
		Hospitalization	4.0305%	7.2258%	15.9505%

**Table 3 vaccines-10-01387-t003:** Base-case one-year influenza disease burden comparisons of the three different QIVs.

	Influenza Cases (No., %)	Influenza-Associated Complications (No.)	Influenza-Associated Hospitalizations (No.)	Influenza-Associated Deaths (No.)
QIV	371,742	16,824	7448	1526
aQIV	336,353	15,222	6739	1381
HD-QIV	343,599	15,550	6884	1410
Difference (aQIV-QIV)	−35,390	−1602	−709	−145
Difference (HD-QIV-QIV)	−28,143	−1274	−564	−116
Difference (aQIV-HD-QIV)	−7247	−328	−145	−30

**Table 4 vaccines-10-01387-t004:** Probabilistic sensitivity analysis results based on 1000 simulation runs.

Outcomes	Aqiv-QIV Mean (95% CrI)	aQIV-HD-QIV Mean (95% CrI)
Mean difference in influenza cases (No.)	−34,171 (−150,413 to −182)	−6945 (−43,812 to 5891)
Mean change in complications cases (No.)	−1546 (−6962 to −9)	−313 (−1954 to 265)
Mean change in influenza-associated hospitalizations (No.)	−685 (−3048 to −4)	−139 (−874 to 121)

aQIV—MF59^®^-adjuvanted quadrivalent influenza vaccine; CrI—credibility interval; HD-QIV—high-dose quadrivalent influenza vaccine; QIV—standard quadrivalent influenza vaccine.

**Table 5 vaccines-10-01387-t005:** Number needed to vaccinate (NNV) for QIV, aQIV, and HD-QIV.

Influenza Subtype	QIV	aQIV	HD-QIV
A/H1N1	136	125	127
A/H3N2	131	91	97
B	56	52	52

aQIV—MF59^®^-adjuvanted quadrivalent influenza vaccine; CI—confidence interval; HD-QIV—high-dose quadrivalent influenza vaccine; QIV—standard quadrivalent influenza vaccine.

## Data Availability

The data presented in this study are available within the article or in the Appendix A.

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
