# Peer review of "A Comparative Analysis of Influenza-Associated Disease Burden with Different Influenza Vaccination Strategies for the Elderly Population in South Korea"

_vaccines, 2022, doi:10.3390/vaccines10091387_

Round 1
Reviewer 1 Report
Study aim to compare different flu vaccination strategies and their effects in elder population in South Korea. Authors suggesting adjuvanted flu vaccine strategy for elderly population might be more wise to reduce the mortalities and disease burden associated with flu.
Study might have some potential merit to scientific field but some major issues were arised from study and needs to be adressed. Please see my comments below,
Line 36: It might be relevant if authors can give excess mortality rates in a range or an average across the world to make a comparison for readers.
Lines 42 to 46: It will be great if you can divide this sentence since it`s difficult to convey a clear message. In addition to that, what authors mean by saying “…8 times the national average of all age groups?
Lines 54 to 62: Please include refs or cite relevant studies here. Besides, ref #3 is not a valid ref, please give a more accurate citation to this report/study.
Fig 1. Words in the boxes are too small or the dpi is not enough, which makes it difficult to read them on the printed copy. Please also avoid using the same or close colors for backgrounds such as “vaccination node repeated”. I recommend authors choose contrast colors when presenting the decision-tree model.
99-101: Does KDCA data sets are available to the public? As you stated published research, please cite those where available. Please could you cite or refer to any published materials for this model input as it`s your material for this study? Raw data as supplementary might work too.
105: Authors might need to state which year of population statistics was given.
116-118: Why results were given under the method section? Do authors need to discuss their findings under methods? Please shift this to the results section and make the discussion under discussion.
Authors expected to provide proper citations for Ref #11. Is it published somewhere that can accessible? Please provide the web address for this ref if it`s not published. Same as for those ref#13 and 14.
Fig. 2. Please use much darker or contrast colors for those boxes; a background color for the boxes might be helpful. If authors didn`t use all the data as shown in fig 2 other than average ones, annual frequencies do not need to be shown here as the authors did not produce any of those given data. In addition to that, any figures, in part or all, should be stated permission by the original producers. Authors must ensure ethical standards for publication.
Table 1 and Table 2. It`s not clear where those data were obtained from. Authors need to address the original ref where data is available.
TableS1. It`s not clear whether the data in the table were originally produced in this study or obtained from somewhere else. Authors need to provide the ref. If data has been obtained from somewhere please cite it. If the data has been calculated in this study which I assume, please shift it under the results.
It`s not clear why authors wanted to show changes in numbers in Table 3 by subtracting them from each other. Authors need to use statistical analysis methods for group comparisons instead of subtracting numbers.
164: Where are the results showing disease burden reduced by 8.19%? Same as for improved by 2.11%? What this all changes indicate for us to convey a clear message? Does it important by significantly?
Figure 3 and 4: Authors expected to explain the Tornado diagram under the method section and what is used for. Fig 3 and 4 do not make sense to readers unless it has been explained clearly.
What are the results in Table 4? It was not mentioned in the text.
I`m not sure the results in Table 5 were interpreted. Authors need to explain how those results are produced and what are they indicating.
Line 214-229: This paragraph has a more relevant place under results unless the authors discuss their findings.
Line 231-42: It`s not clear why authors wanted to address the cost-effectiveness of the vaccines here. It`s not relevant since this study does not compare the costs of vaccines.
Line 252-53: I`m not sure I have understood that meta-analysis of relative effectiveness. What statistical analysis has been performed and found no difference between HD-QIV and aQIV?
I couldn`t locate those findings under the results section.
Line 264: “that” has been repeated, a typo.
Line 259-265: this section should be explained under results clearly thus makes more relevant for us to understand Its importance.
The discussion section is too weak to make an overall justification and importance of their work among the others. The authors did compare their findings with a previously published study (ref#7) which is authored by the same group in part, and it`s not sufficient per my understanding. Previous work based on the cost-effectiveness of vaccine strategies and is different from than current study. In the discussion, there is only one relevant study which authors discussed their findings with, It`s ref#16. Discussion is needs major revision by comparing with more relevant studies. Besides immunosenescence is not related with the submitted study. There is a fact that influenza vaccine effectiveness in elderly population is less when compared to other younger age groups. Immunosenescence is just one of the mechanistic explanations among them that why vaccines less effective in elders, besides it`s not true for all vaccines. When I read the first sentence of the abstract I was expecting to see at least immune response results but cost-effectiveness. Neither cost-effectiveness nor immunosenescence is not relevant to discuss or compare. Authors should give a clearer summary in the abstract, more relevant to their story. Discussion should be more relevant by comparing with others and it must be based on comparing aQIV with QIV or HD-QIV in people who are equal or older than 65 yrs.
Author Response
Please point-by-point responses to the review comments in the attached file.
Thank you.

Reviewer 2 Report
In this manuscript by Yun et al., the authors evaluated different vaccination approaches for seasonal influenza control among adults (≥ 65 years of age) in South Korea, using a one-year decision-tree model to compare influenza disease burdens amongst three study groups vaccinated with three different formulations of a quadrivalent influenza vaccine: a standard dose, a high dose, and an adjuvanted (MF59) dose. Data for the study were sourced from published literature and the Korean disease control and prevention agency. The study indicates that the adjuvanted quadrivalent vaccine formulation is more effective at reducing disease burden among the elderly than a standard dose or a high dose of the vaccine.
Overall, the objective of the study is clear, the analysis was well performed, and the outcome could be useful to policy makers in refining influenza vaccination strategies. It would be interesting to see the outcome of a multi-year analysis (for example, over a decade or even 2 decades) to assess whether the difference in influenza burden among people vaccinated with the standard quadrivalent flu vaccine versus adjuvanted quadrivalent flu vaccine remains similar from flu season to flu season.
Comment
Line 131: The authors assumed a 7.2% rate of influenza among healthy unvaccinated adults despite the unvaccinated adults aged ≥ 65 years constituting an estimated 20% this age group. The authors should provide further justification for the adoption of the method used by Somes et al (reference # 15) in arriving at the assumed 7.2%.
Minor comments
Line 47: change “factors for this is the age-related immunosenescence” to “factors for this is age-related immunosenescence”
Line 52: replace “prevention for” with “preventing”.
Line 66: The authors need to state clearly whether the quadrivalent vaccines used in South Korea are produced by different manufacturers (other than Seqirus).
Line 87: remove “the” before comorbidity.
Line 221: change “For the same the elderly population” to “For the same elderly population”
Line 254: change “nonmaterial” to non-material”
Table S1 should be re-formatted to add the table headers to subsequent pages.
Figures S1 to S6: change “The values in the parathesis” to “The values in parenthesis”
Author Response

(The authors gave the same response as above.)

Reviewer 3 Report
Comments to the author:
The Manuscript " A comparative analysis of influenza-associated disease burden 2
with different influenza vaccination strategies for the elderly 3
population in South Korea”. The manuscript has been written clearly and orderly, and the cited references are updated and generally appropriate to support the sentences. However, I have many doubts about the work done in this study;
Major revisions required
Comments
1. The author said that the influenza infections in South Korea were seasonal but he analysed them for the whole year. Why not studied seasonal comparisons?
2. Immunosenescence is a natural process where the ability of B- Cell and T- cell would be less in differentiating antigenic changes. Adjuvant will prolong the antigen persistence only, not to reduce the immunosenescence. What would be your view on that?
3. Author said that there is comorbidity in connection with the high risk category. One thing: Cardiovascular and pulmonary circulation are inseparable in functioning . How do you differentiate those in terms of influenza infection?
4. Vaccines like aQIV and HD –QIV both are good. Vaccination is prophylactic action to a disease but vaccine is not considered on the basis of cost and your point of introduction and in discussion could be changed.
5. Analytical data period was 2016-2019. Not studied for two years due to pandemic Coronavirus infections. What was the current strategy could also be mentioned in the manuscript.
6. In the line 264 – there is a typographical error, the word “that” has repeated twice.
7. It would be better if more references are included in this manuscript.
Author Response

(The authors gave the same response as above.)

Round 2
Reviewer 1 Report
The MS has been improved.
Reviewer 3 Report
It can be accepted in the present format